# Neuromodulation of Eating Disorders: A Review of Underlying Neural Network Activity and Neuromodulatory Treatments

**DOI:** 10.3390/brainsci14030200

**Published:** 2024-02-22

**Authors:** Kevin Wu, Yu Tung Lo, Jonathon Cavaleri, Matthew Bergosh, Jennifer Ipe, Robert G. Briggs, Kay B. Jann, Stuart B. Murray, Xenos L. Mason, Charles Y. Liu, Darrin J. Lee

**Affiliations:** 1Department of Neurological Surgery, Keck School of Medicine, University of Southern California, Los Angeles, CA 900033, USA; jacklo@nus.edu.sg (Y.T.L.); jonathon.cavaleri@med.usc.edu (J.C.); bergosh@usc.edu (M.B.); ipejennifer@gmail.com (J.I.); gbriggs023@gmail.com (R.G.B.); xenos.mason@med.usc.edu (X.L.M.); cliu@usc.edu (C.Y.L.); darrin.lee@med.usc.edu (D.J.L.); 2Department of Neurosurgery, National Neuroscience Institute, Singapore 308433, Singapore; 3USC Stevens Neuroimaging and Informatics Institute, Keck School of Medicine, University of Southern California, Los Angeles, CA 90033, USA; kay.jann@loni.usc.edu; 4Department of Psychiatry and Behavioral Sciences, Keck School of Medicine, University of Southern California, Los Angeles, CA 90033, USA; stuart.murray@med.usc.edu; 5Department of Neurology, Keck School of Medicine, University of Southern California, Los Angeles, CA 90033, USA; 6USC Neurorestoration Center, Keck School of Medicine, University of Southern California, Los Angeles, CA 90033, USA; 7Department of Biomedical Engineering, University of Southern California, Los Angeles, CA 90033, USA

**Keywords:** eating disorders, anorexia nervosa, bulimia nervosa, binge eating disorder, neuromodulation, deep brain stimulation, transcranial magnetic stimulation, transcranial direct current stimulation

## Abstract

Eating disorders are a group of psychiatric conditions that involve pathological relationships between patients and food. The most prolific of these disorders are anorexia nervosa, bulimia nervosa, and binge eating disorder. The current standard of care involves psychotherapy, pharmacotherapy, and the management of comorbid conditions, with nutritional rehabilitation reserved for severe cases of anorexia nervosa. Unfortunately, many patients often fail to respond, leaving a concerning treatment gap between the current and requisite treatments for eating disorders. To better understand the neurobiology underlying these eating disorders, investigations have been undertaken to characterize the activity of various neural networks, primarily those activated during tasks of executive inhibition, reward processing, and self-reference. Various neuromodulatory techniques have been proposed to stimulate these networks with the goal of improving patients’ BMI and mental health. The aim of this review is to compile a comprehensive summarization of the current literature regarding the underlying neural connectivity of anorexia nervosa, bulimia nervosa, and binge eating disorder as well as the numerous neuromodulatory modalities that have been investigated. Importantly, we aimed to summarize the most significant clinical trials to date as well as to provide an updated assessment of the role of deep brain stimulation, summarizing numerous recently published clinical studies that have greatly contributed to the literature. In this review, we found therapeutic evidence for transcranial magnetic stimulation and transcranial direct current stimulation in treating individuals suffering from anorexia nervosa, bulimia nervosa, and binge eating disorder. We also found significant evidence for the role of deep brain stimulation, particularly as an escalatory therapy option for the those who failed standard therapy. Finally, we hope to provide promising directions for future clinical investigations.

## 1. Introduction

Eating disorders are psychiatric conditions primarily characterized by disturbances in eating behaviors as well as in thoughts and emotions related to eating. Individuals suffering from these disorders experience disordered beliefs related to weight and body image, which can lead to severe psychological and physical harm [1]. The most common of these disorders, and the primary focus of this paper, are anorexia nervosa (AN), bulimia nervosa (BN), and binge eating disorder (BED).

AN is a disorder related to an extreme fear of weight gain and a distorted view of one’s own body, often leading an individual to take extreme measures to maintain or lose weight [2]. The pathologic behaviors exhibited by patients with AN include excessive physical activity, extreme dietary restriction, and purging [2]. Additionally, this disorder can impact an individual’s cognitive and emotional functioning and is often accompanied by medical and psychiatric comorbidities [2], such as bipolar, depressive, and anxiety disorders, as well as life-threatening conditions, including amenorrhea, vital sign abnormalities, malnutrition, the loss of bone mineral density, and abnormal lab findings [1]. AN has a lifetime prevalence of approximately 0.3% [3].

BN involves recurrent episodes of binge eating followed by subsequent inappropriate actions to avoid weight gain from the binge eating episode [4]. These compensatory methods can be harmful and may include self-induced vomiting, laxatives, or prolonged periods of starvation [5]. The psychiatrist Gerald Russell differentiated patients suffering from anorexia nervosa and bulimia nervosa by describing those with BN as normal weight or overweight, while individuals with anorexia nervosa are severely underweight [5]. Sequelae of the disorder due to purging can be fatal and include esophageal tears, gastric rupture, and cardiac arrhythmias [1]. Up to 3% of females and 1% of males will experience BN within their lifetimes [3].

BED is the most common eating disorder [6]. In contrast to bulimia nervosa, BED involves an individual experiencing recurrent episodes of binge eating without subsequent actions to compensate for overeating [4]. The disorder, naturally, is often associated with obesity and metabolic syndrome due to binging and the subsequent high caloric intake without the compensatory measures central to AN and BN [7]. The prevalence of BED worldwide for 2018–2020 is between 0.6 and 1.8% in adult women and 0.3 and0.7% in adult men [8].

The initial management for these eating disorders involves psychotherapy, primarily in the form of cognitive behavioral therapy, the management of psychiatric comorbidities, and nutritional rehabilitation [9,10,11]. Although not standard therapy, medications like antidepressants [12]. And, in the case of AN, antipsychotics are occasionally prescribed [10]. Despite such efforts, a substantial portion of patients are not responsive to standard treatment: 20% of AN patients and 20–26% in BN patients [12,13]. Due to the severity of malnutrition, related adverse events can lead to mortality, most commonly for AN patients at approximately 5% within the first four years [14]. As such, there exists a concerning treatment gap.

Central to the advancement of the understanding of these disorders is the elucidation of the neural network aberrancies underlying them. Among them are the frontostriatal network [15], reward network [16,17,18,19], and default mode network [20,21,22,23,24], which are involved in executive inhibitory control, self-referential processing, reward valuation, and attention. Neuromodulation comprises a collection of modalities that alter brain activation in hopes of alleviating aberrancies noted in these networks. Neuromodulatory methodologies most commonly include transcranial magnetic stimulation (TMS), transcranial direct current stimulation (tDCS), and less commonly deep brain stimulation (DBS).

In this review, we summarize the current literature regarding the neural network correlates of eating disorders. Additionally, we explore the state of research, ranging from case studies to extensive clinical trials, regarding the common neuromodulatory techniques as listed above, as well as two of the lesser explored options in vagal nerve stimulation (VNS) and electroconvulsive therapy (ECT).

In writing this narrative review, searches of the PubMed and Google Scholar databases were conducted with combinations of the following terms: “neuromodulation”, “transcranial magnetic stimulation”, “transcranial direct current stimulation”, “deep brain stimulation”, “eating disorders”, “anorexia nervosa”, “bulimia nervosa”, “binge eating disorder”, “vagal nerve stimulation”, “electroconvulsive therapy”, and “neural networks”. Only articles published in the English language were included. The resulting abstracts were reviewed independently by four authors (K.W., Y.T.L., J.C., and M.B.) for relevance; of these, only primary research conducted in human subjects was included and analyzed. Specifically, we sought to include randomized trials, case series, and should the former study types be unavailable, case reports in order to form a thorough summary of neuromodulation for the treatment of eating disorders. Of note, all relevant randomized controlled trials were included if available due to the weight of their evidence. Additionally, all DBS trials that have concluded in the past 5 years were included as they represent significant additions to the literature since the publication of the most recent reviews. Finally, studies that have not concluded and are currently underway were included for thoroughness.

## 2. Pathological Alterations in Brain Networks

The investigation of brain circuit network underpinnings of eating disorders has revealed a few primary contributing brain regions and networks. Through fMRI, EEG, and structural MRI studies, the interplay between these regions has become more apparent. Here, we examine the primary drivers suspected to contribute to the cognitive state of eating disorder patients and their evident connections.

### 2.1. Frontal–Striatal Circuit

The prefrontal cortex (PFC) has become a particular point of interest with respect to eating disorders as its functions in executive and inhibitory control are believed to be altered in AN, BN, and BED patients [15]. The PFC is the most commonly researched area in studies of eating disorders and is integral to the greater network of the frontal–striatal circuit. This circuit can be partitioned into its dorsal and ventral components, with regions including the PFC, striatum, pallidum, and thalamus [25]. This network, in addition to contributing to the PFC’s functions mentioned above, also plays a role in reinforcement learning and planning [25].

In the context of AN, it is generally agreed upon that the PFC and associated dorsal frontostriatal circuit exhibit overactivity [26,27]. fMRI studies conducted on patients in which AN presented with high- vs. low-fat food picture prompts demonstrated altered connectivity strengths, reinforcing maladaptive behaviors. One study, for example, demonstrated a greater degree of striatum–dlPFC connectivity when viewing the low-fat prompts and a relatively diminished connectivity when viewing the high-fat prompts compared to healthy controls [27]. Conversely, the ventral frontal–striatal network exhibits a heterogenous functional connectivity in the AN state. For instance, in a study by Haynos et al. on resting state functional connectivity (RSFC), patients with AN were found to have lower RSFC between the ventral caudate and frontal regions [28], while studies by Frank et al. and Cowdrey et al. reported the opposite [29,30]. Despite such differences, it is established that the ventral frontal–striatal network, much like the dorsal, exhibits aberrancy in the AN disease state. Further investigation, however, may prove beneficial in elucidating the true behavior of the ventral frontal–striatal states.

In the BN disease state, patients largely exhibit frontal–striatal deficits [22,31]. While in healthy controls, the regions of the frontal–striatal circuit exhibit activation, the same circuit in BN patients exhibit deactivation during conflict-resolution tasks such as the Simon Spatial Incompatibility task [22,31,32]. Cyr et al. additionally found such patterns to exist even in the subthreshold BN population, defined as engaging in at least one binging episode and one compensatory behavior in the past three months, but below frequent enough to reach DSM-V criteria for BN [31]. Altogether, these studies agreed on widespread deactivation across the frontostriatal circuit, with deficits most prominently in the PFC, putamen, striatum, anterior cingulate cortex (ACC), and inferior frontal gyrus [22,31,32]. From a neurocognitive perspective, it is possible that such deficits, reflected in a publication by Marsh et al., represent the inability to resolve conflicts between food intake and weight gain avoidance in the BN disease state, leading to the alternating nature of BN’s binge–purge episodes [22].

Additionally, a paucity of activation in the PFC has been found in the BED state [28,33]. Specifically, when participants in the Veit et al. study were asked to perform a go/no-go task with “healthy” and “non-healthy” food stimuli, patients with BED exhibited a lower degree of PFC activation on functional near-infrared spectroscopy during response inhibition tasks. PFC activation was also significantly negatively correlated with BIS-15 total scores and motor scores [33]. Haynos et al. studied RSFC in BED and also found diminished frontal–striatal network connectivity in comparison to healthy controls, with RSFC decreasing along the dorsal axis. Furthermore, the group discovered a negative correlation between RSFC and binge frequency [28]. As BED is postulated to be a disruption in inhibitory executive control [34], such hypoconnectivity may represent a network corollary for BED’s phenotype.

### 2.2. Reward Networks

The dorsal anterior cingulate cortex (dACC), located within the medial aspect of the frontal lobes, has been associated with the functions of reward value processing, cognitive control, and self-referential processing [16,17]. In studies examining the resting-state activity in patients with AN and BN, the dACC has been found to be synchronously activated with the retrosplenial cortex and the medial orbitofrontal cortex, regions associated with activities of self-referential processing [17]. Interestingly, the synchronous activity varied between disease states: specifically, the activity of the retrosplenial–dACC system corresponding to AN, and activity of the medial orbitofrontal cortex–dACC system corresponding to BN [17]. The authors Lee et al. postulate that such increased synchrony represents the excessive rumination on eating and body shape often seen in AN and BN [17].

In the setting of BED, the dACC demonstrates overactivity [18]. In conjunction with evidence that the ACC’s activity is correlated to food addiction scores [35], greater activation possibly represents a greater affinity for foods, which is common in BED patients. The authors of this study, Geliebter et al., also found heightened concomitant cerebellar activity. As this region is involved in sensory attention and reinforcement learning, the heightened dACC–cerebellar connectivity may represent a circuit hypersensitive to food cues [18].

The orbitofrontal cortex (OFC) has also been demonstrated to play a role in gauging reward value [17]. Lee et al. reported an increased connectivity between the dACC and the OFC in the BN disease state but not the AN disease state, possibly owing to the distinct phenotypic features between AN and BN [17]. Frank et al. further investigated the OFC, finding OFC connectivity with regions such as the hypothalamus to be similarly decreased in both the AN and BN states, suggesting a shared decrease in food reward and appetite [30].

Finally, the subcallosal cingulate cortex (SCC) is postulated to aid in the contextualization of multimodal sensory input and affective information, with a greater response to aversive stimuli. Hayes et al. published the only connectivity analyses focused on the SCC in the context of an eating disorder—AN in this case. As this is an increasingly popular target for DBS (discussed further below), such studies are critical. The group demonstrated SCC activity itself did not differ between or within groups. However, connectivity to the ipsilateral parietal cortex and bilateral thalamus were increased. The authors propose such differences correlate to the increased aversive behaviors, such as rumination or increased response to criticism [19]. Unfortunately, this region has not yet been explored in relation to BN or BED.

### 2.3. Default Mode Network

The default mode network (DMN) is an extensive frontal–parietal network that encompasses the precuneus, dorsal medial prefrontal cortex, posterior cingulate cortex, and angular gyrus [36]. It classically exhibits activity when individuals are occupied by internally driven thoughts or introspection [20,22]. Conversely, the DMN becomes quiescent during the redirection of attention towards an external task or stimuli [22].

In the AN disease state, the DMN demonstrates decreased activity, particularly in the precuneus [20]. Naturally, such lack of activity may reflect an altered degree of self-awareness, particularly surrounding body image [20]. Indeed, when self-images were presented to patients with AN, there was diminished precuneus activation in comparison to controls [37]. In fact, voxel-based structural MRI studies have demonstrated a corresponding decrease in precuneus gray matter volume in patients with AN [38]. Additionally, upon improvement of AN symptomatology, the activity of the DMN strengthens significantly [20,21].

The DMN regions (namely the vmPFC, posterior cingulate cortex, and superior frontal gyrus) exhibit abnormal activity in BN as well [22,23,36]. When asked to respond to cognitive incongruencies (i.e., a Simon Spatial Incongruency task) [22] or perform the Attention Network Task [23], the default mode network failed to deactivate during the tasks. Such failure has been posited to represent the patients’ preoccupation with internally driven thoughts, for instance, body image and food intake [22,23].

Unfortunately, BED lacks the extensive evidence base that has been accumulated compared to AN and BN. One study identified in our review found a reduced level of DMN connectivity in BED compared to both the control and BN groups [24]. The authors propose this relative hypoactivity represents a diminished degree of preoccupation with food or body image thoughts [24], which dictate compensatory behavior in AN and BN [20,22,23].

Overall, evidence for disorder within the DMN in eating disorders is emerging. Unfortunately, this network has yet to be targeted in neuromodulatory techniques although the emerging evidence suggests this may be a promising direction for future research.

## 3. Current Neuromodulatory Options and Their Target Networks/Nodes

### 3.1. Non-Invasive Brain Stimulation

Non-invasive brain stimulation refers to methods of modulating brain and network activity via non-surgical procedures. These methods commonly include repetitive transcranial magnetic stimulation (rTMS) and transcranial direct current stimulation (tDCS). In the context of eating disorders, both methods have been thoroughly studied in vivo, with rTMS representing a larger share of the studies. In the following sections, we will summarize the evidence regarding both methods and their effects on each eating disorder.

Currently, neuromodulatory techniques are neither considered first-line nor approved therapies for eating disorders [39]. This is true for both rTMS and tDCS, the two most commonly investigated non-invasive methods discussed in this review; however, there is substantial evidence to suggest these methods are both safe and beneficial in the treatment of AN, BN, and BED [40,41,42,43,44,45,46,47,48]. In the articles discussed below, the application of rTMS (summarized in Table 1) and tDCS (summarized in Table 2) often followed a prolonged eating disorder course, many times refractory to pharmacotherapy and psychotherapy, which is suggestive of neuromodulation being indicated in cases of resistant eating disorders.

As previously discussed, there are numerous aberrant brain networks involved in disordered eating. Most of the studies of non-invasive brain stimulation have targeted the dlPFC, and several have targeted the dmPFC [32,49]. Likewise, the dmPFC has been shown to have differential activity in AN, BN, and BED when compared to healthy controls [50]. As mentioned prior, the PFC and its partitions play a role in both executive inhibitory control and reward processing, the dysfunction of which is contributory to all of the eating disorders discussed here [51]. Naturally, the targeting of these regions aims to override the aberrancy of the various frontal–striatal networks. To date, no other targets have been investigated, but this is certainly a potential area of study in this nascent field.
brainsci-14-00200-t001_Table 1Table 1TMS studies included in this review, sorted by disease type followed by year of publication.DisorderFirst Author, YearCountry of StudySample SizeTMS PatternNumber of SessionsTMS TargetDuration of ED Initial BMIBMI OutcomeDisease Severity OutcomeAN with comorbid MDDKamolz, 2008 [52]Germany1100 cycles of 10 Hz for 2 s on/10 s off3 series for 26 total sessionsdlPFC4 years12.4 kg/m^2^Increased to 16 kg/m^2^Initial HAMD value of 28 decreased to 11.AN (restricting and binge–purge type) Van den Eynde, 2013 [53]UK1020 cycles of 10 Hz for 5 s on/55 s off1 sessionLeft dlPFC10 (3–30) years15.7 kg/m^2^ (13.8–17.8 kg/m^2^) N/ASensations of “feeling fat” and “feeling full” decreased along with “urge to exercise.”Reduced feelings of anxiety.AN (restrictive with comorbid MDD; binge–purge)McClelland, 2013 [54]UK220 cycles of 10 Hz for 5 s on/55 s off20 sessions; 19 sessionsLeft dlPFC12 years; 35 years15.7 kg/m^2^, 16.4 kg/m^2^BMI decreased at 1 month follow-up in both patients (average decrease of ~0.7 kg/m^2^)EDE and DASS scores decreased in both patients. Patient 1 reported increase in purging frequency. Patient 2 reported decreased purging and laxative use. AN (binge–purge subtype) or BNDunlop, 2015 [55]Canada28 (16 responders)60 cycles of 10 Hz, 5 s on/10 s off20 sessions; 30 for responders with residual symptomsBilateral dmPFC14.75 years19.03 kg/m^2^N/ANo significant difference at baseline between responders and non-responders. Among responders, binge and purge frequency decreased. No change in non-responders.ANMcClelland, 2016 [56]UK60 (49 completed study)20 cycles of 10 Hz for 5 s on/55 s off1 sessionLeft dlPFC9.05 years for TMS group, 11.27 years for sham16.73 kg/m^2^ for TMS group, 16.38 kg/m^2^ for shamN/ASingle session of TMS resulted in lower core AN symptoms of feeling full, urge to restrict, and feeling fat. ANChoudhary, 2017 [57]India11000 pulses of 10 Hz stimulation21 sessionsLeft dlPFC9 years10.94 kg/m^2^17.98 kg/m^2^ at end of 3-week treatment, 18.55 kg/m^2^ at 8-week follow-upLaxative and diuretic abuse decreased significantly.AN (comorbid depression and anxiety)Jaššová, 2018 [58]Czech Republic110 Hz, 15 trains/day, 100 pulses/train, intertrain interval of 107 s 10 sessions Left dlPFC1.5 years12.21 kg/m^2^13.15 kg/m^2^ at discharge, 22.9 kg/m^2^ at 2-year follow-upNo change in Zung self-rating scale (score = 70).ANDalton, 2018 [59]UK3420 cycles of 10 Hz for 5 s on/55 s off20 sessionsLeft dlPFCAverage 14.07 yearsAverage 16.00 kg/m^2^Small but non-significant increases in BMI at end of stimulation and 4-month follow-upSignificant decreases in DASS global score, favoring TMS. ANDalton, 2020 (18-month follow-up from Dalton, 2018) [60]UK3020 cycles of 10 Hz for 5 s on/55 s off20 sessionsLeft dlPFCAverage 14.07 yearsAverage 16.00 kg/m^2^Non-significant increase in BMI at 18-month follow-upHigher rate of weight recovery in TMS group (46% vs. 9%).Non-significant improvements in EDE-Q global in both groups and improvements in DASS-21 were maintained in both groups.ANWoodside, 2021 [41]Canada1910 Hz22.6 average (20–30) sessionsBilateral dmPFCN/A16.4 kg/m^2^ (14.5–18.5 kg/m^2^)Average BMI declined to 16.3 kg/m^2^ at end of treatmentsSignificant improvements in shape concerns and weight concerns in EDE. Additionally, improvement in BAI and BDI.BN with comorbid MDDHausmann, 2004 [61]Austria110 trains of 10 s 20 Hz pulses with a train interval of 60 s10 sessions, twice daily for 5 daysLeft dlPFC9 years18 kg/m^2^N/AAbsence of binge–purge behavior following stimulation treatment. HAMD decreased 50%.BNWalpoth, 2008 [62]Austria1410 trains of 10 s 20 Hz pulses with a train interval of 60 s15 sessionsLeft dlPFCAverage 8.4 years for TMS group, average 8.0 years for sham group.Average 19.6 kg/m^2^ in TMS group, average 19.7 kg/m^2^ in sham group.N/ASignificant improvement in BDI, frequency of binging, and YBOCS at end of treatment, but no significant change between groups. BNVan den Eynde, 2010 [42]UK3820 cycles of 10 Hz for 5 s on/55 s off1 sessionLeft dlPFCMedian 5–10 years in TMS group, median 0–5 years in sham group. Average 25.8 kg/m^2^ in TMS group, average 25.0 kg/m^2^ in sham group.N/ASignificant decrease in urge-to-eat VAS in the TMS group. No significant changes in hunger, urge to binge, mood, tension, or FCQ-S between groups. BNVan den Eynde, 2012 [63]UK720 cycles of 10 Hz for 5 s on/55 s off1 sessionLeft dlPFCMedian 0–5 years in the left-handed group, median 5–10 years in the right-handed group. Average 22.9 kg/m^2^ in left-handed group, average 28.5 kg/m^2^ in right-handed group.N/ANo significant differences in urge to eat, mood, tension, hunger, urge to binge eat, and FCQ-S between left- and right-handed groups. Mood differed significantly between groups, with the left-handed group experiencing a worsening in mood and the right-handed group experiencing an improvement in mood. BN with comorbid MDDDownar, 2012 [64]Canada160 trains of 10 Hz for 5 s on/10 s off20 sessionsBilateral dmPFC28 years20.3 kg/m^2^N/AInitial HAMD of 26 and 28 on the BDI; decreased to 0 at the end of treatment and 7 after 11 sessions, respectively. Binge–purge behavior disappeared completely after session 11 (originally twice-daily 5 h binges with subsequent purging). Single binge–purge episodes on days 65, 70, and 71 post-treatment.BNGay, 2016 [65]France4720 cycles of 10 Hz for 5 s on/55 s off10 sessionsLeft dlPFCAverage 8.0 years in TMS group, average 10.5 years in sham group.N/AN/ANo significant changes in binging, purging, craving, MADRS, or duration of binging. BED with comorbid bipolar II disorderSciortino, 2021 [43]Italy230 Hz bursts at 5 Hz intervals; 2 s on/12.3 s off; 600 pulses per session18 sessions across 3 weeksLeft dlPFC32 years, 10 yearsN/AWeight reduction of 4 kg and 2 kg at 12-week follow-up.HAMD and MADRS improved marginally in both patients. YMRS remained at 0 throughout for both. Complete remission of binging episodes at the end of 2 weeks of treatment. BDI, Beck Depression Inventory; DASS, Depression Anxiety Stress scale; EDE, Eating Disorder Examination; FCQ-S, Food Craving Questionnaire-State; HAMD, Hamilton Depression Rating Scale; MADRS, Montgomery–Asberg Depression Rating Scale; VAS, Visual Analogue Scale; YMRS, Young Mania Rating Scale.
brainsci-14-00200-t002_Table 2Table 2tDCS studies included in this review, sorted by disease type followed by year of publication.DisorderFirst Author, YearCountry of StudySample SizetDCS ParametersNumber of SessionsIntervention TargetDuration of ED Initial BMIBMI OutcomeDisease Severity OutcomeANKhedr, 2014 [66]Egypt7Anodal 2 mA for 25 min with 15 s ramp in and ramp out10 sessionsLeft dlPFC along parasagittal lineMean of 3.4 yearsMean 14.85 kg/m^2^ (12–17 kg/m^2^)N/ASignificant decreases in body dissatisfaction, interpersonal distrust, interoceptive awareness, and ineffectiveness scores of the EDI. Significant improvement in BDI from 22.4 to 13.3.ANCostanzo, 2018 [45]Italy231 mA anodal stimulation18 sessionsAnode over left dlPFC and cathode over right dlPFCN/AMean 14.7 kg/m^2^ for tDCS group, 15.5 kg/m^2^ for sham group. tDCS with “treatment as usual” resulted in significant improvements in BMI. No significant change in family-based therapy with “treatment as usual” group.Significant improvement in multiple eating disorder subscales, but not significant between groups. ANPhillipou, 2019 [67]Australia20Anodal stimulation for 20 min at 2 mA10 sessions Anode over left inferior parietal lobeCurrently underwayANMares, 2020 [68]Czechia130 min of 2 mA anodal stimulation7 sessionsLeft dlPFC anode with cathode over the right orbitofrontal region11 years17.4 kg/m^2^N/ADuring tDCS, the patient developed hyperglycemia and, subsequently, diabetes mellitus.ANUrsumando, 2023 [69]Italy8020 min of 1 mA stimulation1 sessionF3 (anode) and F4 (cathode) of dlPFCCurrently underwayBNKekic, 2017 [46]UK3920 min of 2 mA with 10 s ramp on/off3 sessionsF4 (anode) and F3 (cathode) of dlPFC in one group; F3 (anode) and F4 (cathode) of dlPFC in the other groupMean 9.25 yearsMean of 21.65 kg/m^2^N/ABoth stimulatory groups exhibited decreased self-reported urge to binge eat and increased self-regulatory control.Anode right/cathode left stimulation reduced global MEDCQ-R compared to the other groups. BEDBurgess, 2016 [70]USA3020 min of 2 mA1 sessionAnode on right dlPFC, cathode on left dlPFCN/AMean of 36.1 kg/m^2^N/ASignificant fewer total calories consumed by the tDCS group. Additionally, mean decrease in consumption of preferred foods by 70.28 kcals. tDCS decreased cravings for desserts more than sham. No effect on binge frequency. BEDGordon, 2019 [71]UK662 mA with 10 s fade-out and fade-in6 sessions over 3 weeksAnode on right dlPFC, cathode on left dlPFCCurrently underwayBEDMax, 2021 [72]Germany3120 min of 1 mA (n = 15) or 2 mA (n = 16) 1 sessionAnode over F4, cathode over left deltoid muscleN/AMean 32.1 kg/m^2^ in 1 mA group, mean 33.8 kg/m^2^ in 2 mA groupN/A2 mA group showed significantly fewer binge episodes with 1 mA group showing no changes. 2 mA group demonstrated improved food inhibition in UPPS.BEDGiel, 2023 [73]Germany4115 min of 2 mA6 sessionsAnode over F4, cathode over left deltoid muscleN/AMean 31.9 kg/m^2^ for tDCS + FRIC group, 36.0 kg/m^2^ for sham + FRIC group. Both groups experienced significant reduction in BMIBoth groups experienced significant improvement in EDE and QoL scales. Greater reduction in binge eating frequency in the tDCS + FRIC group vs. sham + FRIC group.BEDFlynn, 2023 [74]UK802 mA10 sessions over 2–3 weeksBilateral dlPFCCurrently underwayBDI, Beck Depression Inventory; EDI, Eating Disorders Inventory; FRIC; Food-related inhibitory control training; MEDCQ-R, Mizes Eating Disorder Cognition Questionnaire-Revised; UPPS, UPPS Impulsive Behavior Scale.


#### 3.1.1. Repetitive Transcranial Magnetic Stimulation (rTMS)

rTMS is non-invasive form of neuromodulation that has demonstrated efficacy in numerous neurologic and psychiatric disorders and is FDA-approved for the treatment of major depressive disorder (MDD), migraine with aura, obsessive compulsive disorder (OCD), smoking cessation, and anxiety with comorbid MDD [75]. In rTMS, local magnetic fields are generated by passing a current through several coils in different orientations around the patient’s head. These magnetic fields can be targeted to regions of interest and induce electrical currents in the brain to modulate neural networks of interest [76]. It is a well-tolerated modality, associated only with simulation site discomfort and headache with rarely reported suicidal ideation and the worsening of psychiatric symptomatology [77].

There are essentially three protocols with which rTMS therapy is delivered: low frequency, high frequency, and theta-burst frequencies. High-frequency stimulation (5–20 Hz) increases cortical excitability and is thought to act through long-term potentiation (LTP), while low-frequency (1 Hz) stimulation is thought to decrease cortical excitability and act instead through long-term depression (LTD) [78,79]. More recently, theta-burst stimulation (TBS) (three pulses of 50 Hz repeated at a 5 Hz frequency) has emerged as a mimic of endogenous brain rhythms [80]. It can be delivered intermittently (iTBS) or continuously (cTBS), whereby iTBS produces excitatory effects, and cTBS produces inhibitory effects [81].

One classical limitation of rTMS is the superficial depth of its effect. To attempt to address this shortcoming, deep transcranial magnetic stimulation (dTMS) was developed as a means of generating magnetic fields that are capable of reaching and influencing deeper brain structures (such as the hippocampus and nucleus accumbens) than that of standard rTMS [82].

##### rTMS for AN

Of all the eating disorders, TMS for AN is the most well-established, with the most common target being the dlPFC [83,84]. The most significant RCT to date is the TIARA study, a double-blind clinical trial of rTMS to the left dlPFC by Dalton and colleagues [59,60]. Thirty-four participants with an illness duration of 3 years or more were randomized to receive either rTMS or a sham [59]. In the TIARA study, not only did participants’ BMI improve, but so did eating disorder symptoms (i.e., feelings of fullness, fatness, and anxiety). Additionally, mood symptoms and quality of life (QoL) were found to be moderately improved. rTMS also demonstrates long-term benefits as the rTMS group demonstrated continued improvement in BMI, mood, and core AN symptomatology at 18 months. In an adjunct study of the same participants, the authors found rTMS resulted in a significant decrease in restrictive, self-controlled food choice behavior [60]. The authors also assessed cerebral blood flow (CBF) via arterial spin label fMRI [85]. Interestingly, while the dlPFC is a critical region in the frontal–striatal regions, the authors found a significant decrease in amygdala CBF in the rTMS group relative to the sham group, with amygdala CBF inversely correlating to BMI. The change in amygdala CBF was proposed to be the result of indirect projections from the dlPFC and suggestive of a reduced fear response regarding weight gain [85].

Smaller studies have largely supported the findings of the TIARA study and the use of TMS for the treatment of AN. McClelland and colleagues performed a randomized, double-blind, sham-controlled trial of rTMS to the left dlPFC in 49 patients [56]. At 24 h post-treatment, they found a tendency toward improvement in core symptoms, although it did not reach statistical significance. Examining changes in core symptoms, McClelland et al. [52] and Van den Eynde et al. [53] reported their experiences with rTMS to the left dlPFC in two and ten patients with refractory AN, respectively, both finding improvements in core symptoms. Case reports by Kamolz et al. [52] and Choudhary et al. [57] also demonstrated significant improvements in BMI, core symptomatology, and in the case of Kamolz et al.’s patient with comorbid MDD, depressive symptoms.

Targeting the dmPFC bilaterally, Woodside and colleagues performed a case series of rTMS in 19 AN patients with comorbid MDD and, in some participants, PTSD [41]. The authors found significant improvements in AN symptomatology and mood symptoms, namely anxiety and depression.

While one case report by Jaššová and colleagues [58] reported no change either in BMI or core symptomatology and Mclelland et al.’s [54] case series of two patients found no improvement in BMI, the existing body of evidence suggests rTMS over an extended timeframe, applied to either the dlPFC or dmPFC, is an effective treatment for AN.

##### rTMS for BN

Second to AN, BN is the most studied eating disorder with respect to rTMS. In regard to the binge–purge behavior characteristic of BN, RCTs exploring rTMS for the treatment of BN were more limited and have conflicting reports of efficacy. Van den Eynde et al. performed a randomized, sham-controlled, double-blind study with 38 patients to investigate rTMS to the left dlPFC [42], finding self-reported food cravings and the frequency of binging episodes to be significantly decreased. However, both Walpoth et al. [62] and Gay et al. [65] conducted RCTs with 14 and 47 participants, respectively. Neither of these two groups were able to find significant differences in binge–purge behaviors between the treatment and sham groups.

When examining psychometrics, the results were equally variable. Walpoth et al. [62] were not able to find significant improvements in the measures of depression or obsessive-compulsive behavior between groups. Conversely, Guillaume et al. investigated the effects of the rTMS treatment on decision making and impulse control [86], finding a statistically significant improvement in impulse control, as well as decision making when assessed using the Iowa gambling task.

There also exist studies of smaller sample sizes that explore rTMS in BN, finding more consistent therapeutic results. In another study by the Van den Eynde group—a case series of rTMS applied to the left dlPFC in seven left-handed patients [63]—the group found decreases in BN core symptomatology with variable changes in mood depending on hand dominance. Hausmann and colleagues’ case study presented a patient with BN and comorbid MDD who underwent left dlPFC rTMS [61], resulting in the complete remission of binging behavior and a 50% decrease in depressive symptoms.

Targeting the bilateral dmPFC, Downar and colleagues reported the case of a patient with BN and comorbid MDD [64]. The patient experienced a complete remission of binging–purging behaviors up until their 2-month follow up. In a subsequent study by the same group, Dunlop and colleagues reported a case series involving 28 patients [55], in which 16 of the 28 patients experienced a >50% reduction in weekly binges at their 4-week follow-ups. fMRI imaging was obtained before and after treatment, showing lower baseline functional connectivity between the dmPFC, and the structures of the lateral OFC and right posterior insula. Responders additionally exhibited lower baseline functional connectivity between the dACC, the right posterior insula, and the right hippocampus, which is consistent with the aforementioned dysfunctional reward pathways. Dunlop and colleagues also identified lower baseline functional connectivity between the dACC and the ventral striatum as well as the anterior insula in responders and found that such connectivity increased with treatment. Conversely, non-responders were found to have high baseline frontostriatal functional connectivity, which was decreased by rTMS and correlated to a worsening of symptoms [55].

Despite promising case studies and case series, multiple sham-controlled, double-blind RCTs failed to observe any difference between rTMS and a sham for patients with BN [62,65]. As such, further RCTs would prove beneficial in characterizing the target, the parameters, and the patient selection that are optimal for rTMS to be effective in BN studies.

##### rTMS for BED

There are few studies addressing rTMS for BED. There is a clear need for more studies to define the role of non-invasive brain stimulation in the treatment of this disorder. Sciortino and colleagues published one of the few studies on the topic, presenting two patients with BED and comorbid treatment-resistant bipolar disorder type II who were treated with iTBS to the left dlPFC [43]. Both patients experienced complete remission in their binge eating symptoms that lasted until their 12-week follow-up visits with only a minor improvement in their depressive symptoms; manic symptoms were absent throughout.

#### 3.1.2. Transcranial Direct Current Stimulation (tDCS)

tDCS is another form of non-invasive brain stimulation that has been studied in a variety of psychiatric and neurologic disorders, including MDD [87], schizophrenia [88], substance use disorder [89], OCD [90], GAD [91], and eating disorders. Although there are currently no FDA-approved protocols for tDCS, it has shown promise in numerous clinical studies [92]. In tDCS, electrodes are placed on the scalp, and a weak current is passed through the brain between the two electrodes. Anodal stimulation is thought to be excitatory, while cathodal stimulation is thought to be inhibitory [93]. Most tDCS protocols deliver a current of 1–2 mA in 10–20 min treatment sessions, and patients undergo 10–20 treatments [92].

##### tDCS for AN

There have been several studies investigating the effects of tDCS on AN, as well as a few ongoing RCTs that are not yet concluded. The largest of the completed studies is a single-blind trial consisting of 23 patients by Costanzo and colleagues. They applied tDCS to the left dlPFC and compared the effects to those of standard therapy [45]. They identified increases in BMI in the tDCS group only at a one-month follow-up. Khedr and colleagues performed an open-label, single-arm study consisting of seven patients who received tDCS to the left dlPFC [66]. In the study, the group found a statistically significant improvement in the core AN and depressive symptomatology at the 1-month follow-up visits. It is worth mentioning a case report published by Mares et al. [68] whereby a patient with comorbid PTSD was discovered to have type I diabetes mellitus (DM) during tDCS to the left dlPFC. It is unclear whether the onset of the patient’s DM was a direct consequence of stimulation.

The currently ongoing studies include a randomized, sham-controlled clinical trial of tDCS to the left inferior parietal lobule (IPL) and is being conducted by Phillipou and colleagues [67] and Ursumando et al. [69]. Phillipou et al. chose the IPL given the decreased functional connectivity between midbrain structures and the IPL that they previously observed in patients with AN [94]. Ursumando et al. [69] are conducting a randomized, sham-controlled clinical trial of tDCS to the left dlPFC in the pediatric population.

##### tDCS for BN

The literature surrounding the tDCS of BN is relatively sparse, with only one primary research article having been identified. That study, conducted by Kekic and colleagues, consisted of a double-blind, sham-controlled proof-of-principle trial utilizing tDCS applied to the right and left dlPFC for 39 patients with BN [46]. In their study, patients received three sessions of tDCS—anode right/cathode left, cathode right/anode right, and sham—in a counterbalanced, randomized order. A variety of binge eating and psychologic tests were performed after each session and at 24 h. Binge eating symptoms and the ability to value delayed rewards improved with both the right and left anodal montages. Interestingly, mood symptoms improved only with the right anodal montage. Such results signify the importance of further exploration of tDCS’s role in the treatment of BN [46].

##### tDCS for BED

The literature regarding tDCS for the treatment of BED is more numerous. Overall, the results have been consistently promising.

In a case series involving 30 patients with BED, Burgess and colleagues performed 2 mA tDCS to the dlPFC or a sham in a counterbalanced study paradigm [70]. They found tDCS to be associated with decreased food cravings and intake, as well as binge eating desire (with a larger effect size in the male cohort). In a double-blind RCT, Max et al. investigated the effect of tDCS to the dlPFC in a counterbalanced order in 31 patients [72]. They performed a food-modified antisaccade learning task and found that the 2 mA tDCS protocol decreased latencies and binge eating episodes, whereas the 1 mA tDCS increased latencies and had no effect on binge eating episodes. These results suggest that tDCS monotherapy at 2 mA may improve eating disorder symptoms, possibly as a result of improved inhibition functions regarding rewarding food stimuli, while 1 mA stimulation may have a detrimental effect.

Additionally, there is considerable interest in studying tDCS as an adjunct therapy for BED. In a phase-II sham-controlled, double-blind randomized control trial, Giel and colleagues assessed the effect of combining inhibitory control training with right-anodal 2 mA tDCS to the dlPFC vs. a sham in 41 patients with BED [8]. They found a significant reduction in binge eating frequency in both the treatment and the sham groups at 4 weeks and 12 weeks, with a statistically significant difference between the treatment and sham groups only at the 12-week timepoint.

Gordon et al. are currently conducting a sham-controlled crossover RCT that examines the effect of a combined approach bias modification (ABM) training co-administered with either anodal tDCS to the right dlPFC or a sham in 66 patients with BED [71]. ABM is a learning method that reinforces avoidance behavior in response to food cues. The authors of the study have thus far published on the patients’ experience with ABM, and overall, patients have found ABM to be a worthwhile activity [95]. Full results from the work by the group are pending. The TANDEM trial by Flynn and colleagues is another exciting single-blind, sham-controlled randomized trial that will look at self-administered at-home anodal tDCS to the right dlPFC with concurrent ABM training [74].

While we are awaiting results from two adjunctive tDCS studies, the current released body of evidence suggests that 2 mA tDCS may be an effective non-invasive option for the treatment of BED.

#### 3.1.3. Electroconvulsive Therapy (ECT)

ECT is the oldest form of non-invasive brain stimulation that has shown some efficacy in a variety of psychiatric disorders, including treatment-refractory depression [96], bipolar disorder [97], and schizophrenia [98]. ECT induces a brief, generalized tonic-clonic seizure through an external current [99]. It is generally a well-tolerated procedure with a low risk profile, and its use is mostly restricted to the acute inpatient treatment of severe, refractory psychiatric disorders (e.g., mania, schizophrenia, major depressive disorder, and suicidality) [97,100]. Despite its well-established use in multiple psychiatric conditions, there have been few studies investigating the efficacy of ECT in the context of eating disorders.

A systematic review by Pacilio et al. included 11 studies with 14 patients (13 with AN and 1 with BED) with a wide age range (12–94), number of treatments (5–22), and variety of comorbid conditions (MDD, anxiety, OCD, psychosis, and suicidal ideation/attempts) [100]. In total, 13 of the 14 patients showed some improvement in their disordered eating with no serious adverse events. Another systematic review by Andersson et al. (2023) looked specifically into AN with comorbid severe depression [84]. This review included 46 patients with a large age range (12–94), number of treatments (5–31), and a variety of comorbid psychiatric disorders. Of note, 45 of the 56 patients were female, and only 3 of the 46 patients fulfilled the criteria for severe AN. In terms of outcomes, they observed a median increase in BMI of 0.4 kg/m^−2^, and they saw a 50% reduction in the Montgomery–Asberg Depression Rating Scale (MADRS). Taken altogether, ECT has shown some efficacy in treating the core symptomatology of eating disorders as well as mood symptoms.

Given its restricted use to acute psychiatric crises and the need for general anesthesia, ECT is unlikely to find significant use in the outpatient management of disordered eating when compared to other non-invasive brain stimulation methods.

### 3.2. Invasive Neuromodulation

As opposed to non-invasive brain stimulation, invasive neuromodulation involves surgical intervention and the placement of deeper-reaching stimulation devices. While these methods include both deep brain stimulation (DBS) and Vagal nerve stimulation (VNS), the vast majority of invasive neuromodulation for eating disorders involve DBS. In the following sections, we will summarize the evidence regarding both their methods and effects on each eating disorder.

The curation of potential targets for DBS largely stems from attempts to treat comorbid conditions, namely MDD [101] and OCD [102]. Today, studies primarily target the subcallosal cingulate (SCC) target [101,103] and the nucleus accumbens (Nacc) [104,105,106], the exploration of which have been largely led by Canadian and Chinese groups, respectively. Alternatively, Israël et al. reported one of the earliest cases of SCC DBS in a 52-year-old woman with comorbid depression [101] as the SCC is a target that was previously explored for the treatment of depression as well as OCD. Furthermore, studies of OCD patients who underwent Nacc DBS also reported improvement in anorexia symptoms [102].

#### 3.2.1. Deep Brain Stimulation

DBS involves the implantation of an electrode into a deep brain structure [104]. Given this degree of invasiveness, DBS is generally an escalatory treatment option, usually reserved for those that are refractory to less invasive methods [103,104,107]. The Nacc and subcallosal cingulate (SCC) are the most common targets for DBS (Figure 1). The evidence is largely limited to a handful of case series and reports, which are summarized in Table 3. Many studies argued that DBS is warranted if patients are in a life-threatening situation due to a medically refractory disease [104] with no other reasonable treatment options, as the potential benefits would then outweigh the risks of surgery.

Most studies defined surgical candidacy through a combination of severity and duration, although the threshold for intervention varied widely between studies. Generally, there should be a balance between allowing adequate time for natural history and/or response to medications, and the enabling treatment for patients who remain medically unstable with just non-surgical treatment alone. Earlier studies by Wu et al. [105] and Wang et al. [106], on the other hand, required disease durations of 1 to 2 years prior to the implantation of DBS electrodes.

Investigating the SCC, Lipsman et al. reported their experience with DBS implanted into the SCC for AN, with three out of six participants achieving an increase in BMI from the baseline at a nine-month follow-up [103,110]. For the initial three months post-surgery, there was a reported decrease in BMI, which the authors attributed to a regression to the baseline, as these patients all underwent a period of inpatient treatment for preoperative optimization prior to DBS implantation. This was followed by a gradual increase in weight that was sustained for more than nine months post-DBS implantation. A separate study by the same group followed 16 patients under the same protocol with a follow-up after up to one year. Again, Lipsman et al. found that patients who underwent bilateral SCC DBS experienced significant increases in BMI and significant improvements in various psychometric measures (full battery of measures in Table 3) [110]. Finally, recruiting from the patients in these previous studies, the group was able to follow 15 patients over the course of 3 years. Once again, the group was able to demonstrate an increase in the mean BMI (from 14.0 kg/m^2^ to 16.3 kg/m^2^), although the mean monthly binging/purging frequency was unchanged compared to the baseline [115]. Importantly, with regard to non-primary measures, the mean QoL did not improve, but the employment rate increased.

Regarding the Nacc, evidence consistently reports improvements in BMI. Wu et al. contemporarily implanted bilateral DBS into the Nacc of four patients with refractory AN, a subcomponent of the ventral striatum and, as mentioned prior, a crucial part of the reward circuitry. Here, the team found significant increases in BMI [105]. Importantly, in comparison to Lipsman et al.’s study, Wu et al. reported on a patient population with a significantly shorter duration of disease (13 to 28 months, versus 4 to 37 years in the Lipsman et al. study [103]). It is possible the BMI increase could be the result of secondary pharmacotherapy, as these patients also received concurrent SSRIs and olanzapine. The same group, Liu et al., published results from a larger cohort in 2020, with similar DBS parameters, resulting in 12 out of 28 (43%) patients achieving a BMI of >18.5 kg/m^2^ at 2 years [104]. The group also discovered via post hoc analysis that the restrictive AN subtype responded better than the binge eating subtype. In a case series conducted by Wang et al. in 2013, the team recruited a combined cohort of six patients who underwent bilateral Nacc RF ablation and two patients who underwent bilateral Nacc-DBS; in both groups of patients, a significant BMI increase was observed by one year, as well as significant improvements in QoL, anxiety, OCD, and depression measures [106]. In the only study to report no improvement from DBS to the Nacc, the Oxford group of Scaife et al. (2022) implanted DBS electrodes in seven patients [116]. While three of the patients experienced greater than or equal to a 35% increase in the Eating Disorder Examination (EDE) scores, no significant changes in BMI were found. In the only study discovered during this review regarding the use of DBS for the treatment of BED in humans, Shivacharan et al. (2022) applied bilateral Nacc DBS in two patients with the aim of treating their obesity that was secondary to BED. The group found improvement in self-control metrics relating to food intake and a decrease in BMI in both patients of 4.5% to 5.8% at 6 months [107].

Martínez et al. examined both targets by implanting DBS electrodes in either the SCC or Nacc, depending upon comorbidities (affective or anxiety disorder, respectively), in eight patients with refractory AN [114,117]. For those without a predominant comorbidity, the binge–purge subtype of AN received DBS to the Nacc, and the restrictive subtype of AN received DBS to the SCC. At 6 months, five of the eight participants experienced a ≥10% increase in BMI compared to the baseline with significant improvements in QoL measures.

DBS targeting the bed nucleus of the stria terminalis (BNST)—a target typically studied in the context of OCD—is far less common, and results from published studies have been conflicting [111,112]. It is believed that the BNST forms a network with the lateral hypothalamus and Nacc—regions involved in feeding and reward processing [118]. Blomstedt et al. first reported on the case of a 60-year-old patient with AN and comorbid MDD who received medial forebrain bundle (MFB) stimulation followed by bed nucleus of the stria terminalis stimulation (BNSF). Under the MFB condition, the greatest improvements were seen in the MADRS, Hamilton Anxiety Index, and the Global Assessment of Function Scale. The patient, however, experienced blurred vision as a complication of MFB stimulation, and was re-operated on to implant BNSF DBS electrodes. Under her new condition, depression ratings improved, as did anxiety and global function ratings. However, under both conditions, BMI changes were marginal (<0.3 kg/m^2^) [111]. Two years later, Manuelli et al. published their experience with a 37-year-old female patient diagnosed with severe restricting-type AN. Bilateral DBS was applied to the BNST, resulting in an improvement in bulimic, body uneasiness, and QoL scores. Importantly, the patient’s BMI increased at nearly every monthly follow-up for the duration of the study (6 months) [112].

The current body of evidence suggests that the SCC and Nacc may be effective DBS targets in the treatment of AN. Conversely, the evidence for the targeting of the BNST and the treatment of BED is less convincing due to conflicting results and a lack of studies, respectively. Moreover, there is a notable lack of evidence regarding DBS in the treatment of BN. Additionally, it should be noted that while most studies reported significant weight restoration and improvement in ratings of comorbid symptoms (i.e., depression, OCD, and anxiety), QoL improvement is less certain. Most studies did not provide strict metrics for QoL, although it is worth noting that among those that did [103,106,116,117], all but De Vloo et al. found improvements in such measures [115]. In reference to their findings, De Vloo et al. suggested that an early post-operative placebo effect and expectation bias could contribute to more positive short-term results which diminish in the long term [115]. Importantly, there is no guarantee that weight recovery implies improvement from the underlying psychological cause of anorexia nervosa. Future investigations can aid in this research by including QoL measures in their battery of tests and surveys.

#### 3.2.2. Vagal Nerve Stimulation (VNS)

Vagal nerve stimulation is currently FDA-approved for major depressive disorder and epilepsy; however, there is evidence for VNS as a treatment for eating disorders [119,120,121]. In the studies that followed, vagal dysregulation was shown to exist in the bulimia nervosa, anorexia nervosa, and binge eating disorder disease states [122]. Despite preclinical and clinical investigations, the specific mechanism by which VNS normalizes behavioral and neural processes regarding eating is still unclear.

In our review of the literature, we identified one article detailing the use of VNS to specifically target AN, BN, and BED symptomatology, which was written by Melis et al. They utilized transauricular VNS in 15 patients with eating and feeding disorders (applied to the left ear). They reported changes in anxiety (via BAI) and depression (via HAM-D) [123]; however, they did not report BMI outcome; 84% of participants experienced the remission of anxiety, while 79% experienced the remission of depression. Overall, such findings suggest VNS successfully inhibits parasympathetic hyperactivation in response to varied emotional states [124,125].

## 4. Discussion

In this review, we offer an extensive and up-to-date summary of the current literature regarding neuromodulation in multiple eating disorders and underlying their network aberrancy. Specifically, we collated literature from invasive neuromodulation (i.e., DBS and VNS) and non-invasive neuromodulation (i.e., rTMS, ECT, and tDCS) as well as explored the three most commonly implicated networks: the frontostriatal network, reward networks, and the default mode network.

While past reviews have discussed similar topics to those covered in our review—for example, Gallop et al. published a review regarding the various neuromodulation options available [49], and Stice’s group published a network review with a particular emphasis on functional near-infrared spectroscopy and real-time fMRI [126]—we aimed to incorporate the multiple focuses of these well-written reviews. In doing so, we included a survey of the various brain networks implicated in eating disorders, and how different neuromodulation techniques influence these networks. Additionally, since the publication of many of these reviews, a number of clinical trials have concluded, particularly regarding the use of DBS in the treatment of various eating disorders with an emphasis on AN [104,107,111,112,113,114,115]. We believe it is particularly pertinent to discuss these newly published studies as they have come to represent a substantial portion of the available literature.

### 4.1. Effects on Brain Networks/States

The literature currently available demonstrates aberrancies within multiple networks, with the most commonly discussed being the frontal–striatal networks, reward networks, and default mode network.

Neuromodulation largely targeted the frontostriatal network due to the (1) implication of the PFC in multiple psychiatric illnesses [37,127,128,129] and (2) the accessibility of the PFC [130], specifically for the localized non-invasive modality of rTMS. This network notably includes the PFC, putamen, striatum, ACC, and inferior frontal gyrus [22,31,32]. Indeed, the investigation of PFC rTMS’s effect on eating disorders is the most extensive, making use of the region’s role in executive inhibitory control. As mentioned previously, there is disagreement with the function of the ventral frontal–striatal circuit in the AN state; this would benefit from further investigation.

Furthermore, there is evidence for the importance of the OFC, a region central to the cortical reward networks [50,131,132], in eating disorders. The OFC exhibits increased functional connectivity to the ACC and hypothalamus in the BN and AN disease states, respectively [17,30], and to our knowledge, no such investigation has been undertaken to date. As the region plays a significant role in the reward valuation process, its modulation may assist patients with obesity in resolving their overvaluation of food stimuli [17]. Of note, the Devoto group has demonstrated increased functional connectivity between the OFC and the whole brain following PFC dTMS for the treatment of obesity [131]. Such results may suggest a role for OFC targeting in the treatment of other eating disorders, such as AN, BN, and BED, the latter of which is most closely linked to obesity [133,134]. There is also evidence for SCC and DMN as neuromodulation targets. The aberrance in the SCC’s functional connectivity is an area of great interest as the region is a promising target for DBS [101,103,114]. While investigation into the DMN’s role in eating disorder pathogenesis has been more thorough than that of the SCC, there are still notably fewer than those of the frontal–striatal and reward networks.

Finally, multiple studies have examined the state of networks following recovery from eating disorders. Interestingly, the studies identified in this review found preserved or more exaggerated abnormalities in functional connectivity and activation patterns. For instance, in the context of AN, dlPFC activation is greater in recovered patients [135]. Similarly, in response to rewarding food stimuli, recovered AN patients continue to exhibit greater activation responses in the ventral striatum and caudate [29,136]. In recovered BN patients, a partial reversal of striatal activation deficits has been identified [137], and for PFC activation deficits, it was determined that cerebral blood flow in the PFC was inversely related to duration of recovery [30]. The authors have proposed such persistent activation patterns are suggestive of ongoing increased salience attribution to reward food cues [29], enhanced self-regulatory processes [138], and difficulty distinguishing between positive and negative feedback cues [139]. No such studies exist regarding the treatment of BED.

As recovery becomes more chronic in eating disorder patients, functional connectivity and activation patterns could greatly add to the knowledgebase. Specifically, it would be valuable to understand the evolution of network aberrancy and determine if the aforementioned findings are sustained or reversed.

### 4.2. Neuromodulation as an Emerging Treatment Modality

Despite the application of psychotherapy, the treatment of psychiatric comorbidities, and nutritional rehabilitation [9,10,11], it is estimated that up to a fifth of AN patients [13] and a quarter of BN patients [12] do not respond to the current standard of care. Various neuromodulation techniques have been proposed to bridge the existing treatment gap, with the aim of reducing such serious morbidities and mortalities. DBS in particular has shown promise in preclinical studies in reducing certain illness behaviors and enabling weight gain (an average of a 14% increase in AN, based on a meta-analysis by Karaszewskaet et al. [13], and up to a 50% increase in BMI in some DBS studies [105,113]), and several groups—particularly those utilizing DBS, rTMS, and tDCS—have demonstrated such benefits in early-phase clinical trials [59,60,103,110,115,138].

The primary outcomes commonly measured include BMI changes or changes in binge–purge frequency. However, it is less clear how these primary measures translate to QoL gains. QoL outcomes were generally improved across the rTMS, tDCS, and DBS modalities, with one group reporting no benefit [115]. However, a large portion of the included studies, including a majority of studies focused on DBS, did not include QoL measures. There exists complexity and variability between psychosomatic interactions in ED and resolving the behavioral aspects (i.e., increased food intake in the form of BMI increase or reduction in binging–purging episodes) may not equate to improved psychological aspects of ED (e.g., dysmorphic body perception). As such, future studies should further evaluate the impact on quality of life with these neuromodulation methods.

### 4.3. Considerations for Invasive Neuromodulation

The inherent surgical risks involved in invasive procedures such as DBS limit them to patients who are refractory to non-surgical treatments and for whom the benefits outweigh the surgical risks. Here, we discovered that surgical indications for the procedure vary widely among groups, lacking a universal disease symptom severity/duration threshold for surgery [104]. However, given that eating disorders can be resistant to psychotherapy and pharmacotherapy, DBS for the treatment of eating disorders should only be explored as an adjunctive or escalatory treatment option. In our review, we identified multiple studies assessing the efficacy of DBS for the treatment of AN. While there exists variability in results, with some studies reporting a stable BMI [101,114,116], the vast majority reported improvements in BMI, with none reporting a decline (Table 3). Notably, we did not identify studies utilizing DBS for the treatment of BN, representing a potential future direction for investigation. Likewise, the majority of DBS studies included in this review did not aim to investigate the efficacy of DBS in those who failed a less invasive neuromodulatory therapy. As such, future investigation may aid in elucidating DBS’s role as an escalation of non-invasive neuromodulation by focusing on patients who have failed TMS or tDCS.

Additionally, in appraising the use of DBS for eating disorders, ethical considerations have been raised. In evaluating these risks in the treatment of AN, Park et al. and Maslen et al. raised several shared concerns, namely potentially questionable patient consent, a lack of understanding of the disease’s pathology, a continued need for thorough outcome measures, and general DBS safety [139,140]. As severe AN exacerbation may be met with nutritional rehabilitation without patient consent, both author groups expressed concerns over the possibility that DBS may be administered similarly. Moreover, the validity of consent may be a point of concern as patients with AN can exhibit altered decision making in the goal of minimizing weight gain. Alternatively, an element of coercion may exist as patients may seek DBS as a means of avoiding nutritional rehabilitation. In aiming to reduce these risks, Park et al. recommended the inclusion of an independent qualified clinical ethicist and/or clinician to assess capacity and independently advocate for the patient. Additionally, as discussed in our review, the effects of DBS on the psychological effects of AN have been less thoroughly investigated; this uncertainty has been echoed by Park et al. and Maslen et al. The uncertain underpinnings of AN may provide a further point of concern as DBS may lead to unintended changes in mood or behavior that complicate future or continued consent for treatment. The authors also share a concern regarding the common risks associated with intracranial operations (i.e., seizures and hemorrhage) [139,140].

Cost analysis studies utilizing quality-adjusted life years (QALY) may prove beneficial for the future application of invasive neuromodulation in eating disorders. Such a study does not currently exist; however, those conducted for DBS in Parkinson’s disease [141], Alzheimer’s disease [142], and major depressive disorder [143] have demonstrated consistent long-term cost savings. Given that the productivity cost of eating disorders has been estimated to be over USD 326 B per QALY [144], DBS may prove economically preferable.

### 4.4. Limitations of the Current Study

It should be noted that as a narrative review, the literature search conducted is not as encompassing as that of a systematic review or meta-analysis. As such, there is a naturally increased risk for selection bias. To address this, in writing this review, we aimed to include articles of high evidence quality or of therapeutic intervention novelty (i.e., unique applications of neuromodulation) to offer a summary of the broad and emerging field. Furthermore, literature searches and reviews were conducted by four authors (K.W., Y.T.L., J.C., and M.B.), independently.

It is also worth noting that there exists potential publication bias in the included literature, as outcomes supporting the null hypothesis may be less often shared. Additionally, many of the included studies had low sample sizes, resulting in an overall lower quality of evidence. This is particularly evident in the DBS literature as participants are often of high acuity, and the administration of the therapy is relatively more invasive—both of which are significant factors limiting patient recruitment. Finally, a majority of the included literature (with the exception of RCTs) may benefit from the inclusion of a sham group to minimize the effect of the placebo.

As such, future studies may aid in reducing bias by incorporating an RCT design and expanding sample sizes.

## 5. Conclusions

In conclusion, there exists considerable evidence supporting the therapeutic benefits of multiple non-invasive and invasive neuromodulatory therapies for eating disorders. However, considerable future evidence is needed to clarify variable results and better understand the network changes that accompany eating disorder development and recovery.

## Figures and Tables

**Figure 1 brainsci-14-00200-f001:**
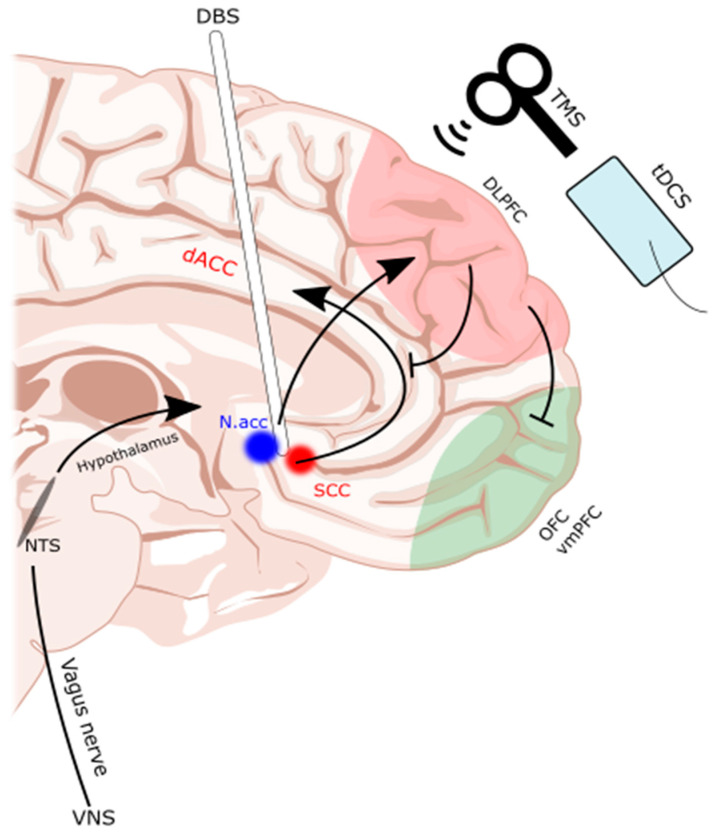
Subcallosal cingulate and nucleus accumbens as common DBS targets. TMS, tDCS, and DBS interventions are illustrated on a midline sagittal brain with the dorsolateral prefrontal cortex (dlPFC) and orbitofrontal cortex (OFC)/ventromedial prefrontal cortex (vmPFC) of the frontostriatal networks highlighted. Additionally, the reward pathways of the deep brain are highlighted as they were targeted by the DBS electrode. dACC: dorsal anterior cingulate cortex. DBS: deep brain stimulation. Nacc: nucleus accumbens. NTS: nucleus tractus solitarius. SCC: subcallosal cingulate. tDCS: transcranial direct current stimulation. TMS: transcranial magnetic stimulation. VNS: vagal nerve stimulator. Illustration based on brain atlas by Schaltenbrand and Wahren, 1977 [108].

**Table 3 brainsci-14-00200-t003:** Summary of included DBS cases, arranged by chronological order of study dates.

First Author, Year	Country of Study	Sample Size	Intervention Target	Stimulation Parameters	Disorder	Inclusion Criteria	BMI Criteria	BMI Outcome	Disease Severity Outcome
Israël, 2010 [101]	Canada	1	Bilateral SCC	Unilateral stimulation on right side at 130 Hz, 5 mA for 2 min on, 1 min off	Restrictive AN	N/A. Patient has a disease duration of 35 years	N/A	Stable BMI at 2 years (19.1)	EAT-26 was 1.04 and 1 at 2 and 3 years. Low EDE score at 3 years “comparable to normal population”. No QoL score.
Barbier, 2011 [109]	Belgium	1	Bilateral ALIC, BNST	N/A	AN with comorbid OCD	N/A. Patient has a disease duration of 24 years	N/A	From an initial BMI of 13.1 kg/m^2^, BMI increased to 13.7 kg/m^2^ and 23.0 kg/m^2^ at 2-week and 3 month follow-up, respectively	The patient exhibited reduction in YBOCS, EDE, EDI, food phobia survey, MADRS, and an increase in global function scores.
Lipsman, 2013 [103]	Canada	6	Bilateral SCC	Bilateral stimulation at 130 Hz and 5–7 volts with a pulse width of 90 μs	Restricting or binge–purge AN	Inclusion criteria: >2 years if increasingly medically unstable >3 years if relentless unresponsive >10 years if stable Actual: 4–37 years duration	≥13 kg/m^2^	50% (3 of 6) patients had higher BMI at 9 months than baseline	YBC-EDS (preoccupations) changed from 23.7 preoperation to 17.7 at 6 months. YBC-EDS (rituals) changed from 29.3 to 19.0. Decreases in HAMD, BDI, YBOCS, and BAI. QoL score increased in those who gained weight.
Lipsman, 2017 [110]	Canada	16	Bilateral SCC	Bilateral stimulation at 130 Hz and 5–7 volts with a pulse width of 90 μs	Restricting or binge–purge AN	Inclusion criteria: >2 years if increasingly medically unstable >3 years if relentless unresponsive >10 years if stable Actual: 4–37 years duration	≥13 kg/m^2^	BMI improved from 13.83 kg/m^2^ at baseline to 17.34 kg/m^2^	Significant improvement in HAMD, BAI, and DERS at 12-month follow-up. No QoL measures reported.
Wu, 2013 [105]	China	4	Bilateral Nacc	N/A	AN (subtype not specified)	>12 months duration of illness (range: 13–28 months)	None specified (range: 10–13.3)	Average 65% increase in BMI at 38-month follow-up	No AN-specific assessments reported. YBOCS and HAMA scores were reduced on average. No QoL measures were reported.
Wang, 2013 [106]	China	2 (6 more underwent RF ablation)	Bilateral Nacc	Bilateral stimulation at 135–185 Hz and 2.5 to 3.8 volts with a pulse width of 120–210 μs	AN	>2 years duration of illness	Not specified	Both DBS patients experienced increases in BMI from 13.1 kg/m^2^ to 18.0 kg/m^2^ and 12.9 kg/m^2^ to 20.8 kg/m^2^, respectively, at 1-year follow-up	No ED-specific scale. QoL: SF-36 improved in physical functioning, role-physical, bodily pain, social functioning, and role-emotional 1 year post-operation. General health, vitality, and mental health were improved at 6 months and 1 year post-operation. Social functioning: SDSS scores improved.
Blomstedt, 2017 [111]	Sweden	1	Bilateral medial forebrain bundle, followed by bilateral bed nucleus of the stria terminalis	Bilateral medial forebrain bundle stimulation at 130 Hz and 2.8 to 3.0 volts with a pulse width of 60 μs Bilaterally bed nucleus stimulation at 130 Hz and 4.3 volts with a pulse width of 120 μs	AN and comorbid MDD	N/A. Duration of disease not specified. Originated during childhood for this 60-year-old patient	N/A	BMI marginally increased at 12 months from 16.2 kg/m^2^ to 16.5 kg/m^2^ under medial forebrain bundle stimulation. BMI marginally decreased from 14.5 kg/m^2^ to 14.3 kg/m^2^ under bed nucleus stimulation	Medial forebrain bundle stimulation improved MADRS, HAMA, and GAF scales, but worsened HAMD. Bed nucleus stimulation resulted in improvement in HAMD and GAF, with marginal improvement in MADRS, and worsening of HAMA.
Manuelli, 2019 [112]	Italy	1	Bilateral bed nucleus of the stria terminalis	Bilateral stimulation at 130 Hz and 4 volts with a pulse width of 60 μs	AN	N/A. Patient has a disease duration of 18 years	N/A	BMI steadily increased monthly from an initial 16.31 kg/m^2^ to 18.98 kg/m^2^ at 6 months	Consistent improvement in BUT subscores, except for “depersonalization”, which showed variable changes. BITE, EAT-26, and YBOCS scores also consistently improved. QoL: SF-36 showed consistent monthly improvement.
Liu, 2020 [104]	China	28	Bilateral Nacc	Bilateral stimulation at 160–180 Hz and 2.5–4.0 V with a pulse width of 120–150 μs	AN	>3 year duration of illness, resistance to medical treatment for at least 3 months	Not specified	BMI significantly improved from baseline of 13.01 kg/m^2^ to 15.29 kg/m^2^ and 17.73 kg/m^2^ at 6-month and 2-year follow-ups, respectively	No ED-specific scale. Significant decreases in YBOCS, HAMA, and HAMD at 6 months and 2 years. Significant increase in MMSE at 6 months and 2 years. Social functioning: SDSS improved from 11.14 to 8.64 at 6 months and 4.22 at 2 years after surgery. No QoL measures reported.
Arroteia, 2020 [113]	Luxembourg	1	Bilateral Nacc	Bilateral stimulation at 204 Hz and 4.5 to 5.5 mA with a pulse width of 350 μs	Bulimic AN	Not specified	Not specified. Patient’s BMI 12.8 kg/m^2^	46.9% increase in weight at 12-month follow-up. At 14 months, binge eating and purging frequency increased, which persisted until 19 months. DBS explanted at 24 months due to infection	No ED-specific scale. No QoL or social functioning outcomes. Patient subjectively reported no change in behavior (anorexia/bulimia) but reported improvements in mood and energy.
Villalba Martínez, 2020 [114]	Spain	8	Bilateral Nacc or SCC	Bilateral stimulation at 130 Hz with a pulse width of 90 µs Amplitude started at 3.5 mA and increased per patient tolerance	AN	Age of 18–60 years, >10 years duration of illness, and refractory to treatment (no response to ≥3 voluntary intensive treatments or clinical deterioration and rejection of further treatment with ≥2 involuntary hospital for nutritional rehabilitation)	≥13 kg/m^2^. One patient presented with a lower BMI and received preoperative admission for optimization of BMI	No change in mean BMI at 6 months. However, when adjusting for need for preoperative optimization, there was revealed to be a ≥10% increase in BMI in 5 patients	Mean increases in SF-36 scores (QoL measure).
De Vloo, 2021 [115]	Canada	15	Bilateral SCC	Bilateral stimulation at 130 Hz and 5.0–7.0 V with a pulse width of 90 µs	Restricting or binge–purge AN	Inclusion criteria: >2 years if increasingly medically unstable >3 years if relentless unresponsive >10 years if stable Actual: 4–37 years duration	≥13 kg/m^2^	Mean BMI increased significantly from 14.kg/m^2^ to 17.5 kg/m^2^ and 16.3 kg/m^2^ at 1- and 3-year follow-ups, respectively	Significant improvements in YBOCS, YBC-EDS, HAMD, BDI, and BAI. No improvement in QoL.
Scaife, 2022 [116]	UK	7	Bilateral Nacc	Bilateral stimulation at 130 Hz and 3.5 to 4.5 volts	AN	>7 years duration of illness. Mean of 21 years (range 12–40 years)	BMI 13–16 kg/m^2^	No significant change in BMI (15.2 kg/m^2^ to 15.3 kg/m^2^) at 12 months. 3/7 patients responded (defined as >35% increase in EDE)	At 12 months, mean EDE reduced from 4.2 to 3.4, (19.0% reduction), mean YBC-EDS reduced from 21.9 to 19.7 (10.0% reduction), and mean CIA reduced from 39.0 to 31.1 (20.3% reduction). HAMD and HAMA also decreased with an increase in SHAPS. QoL: WHO-QoL-Psych improved from 7.9 to 9.4 (18.9% increase).
Shivacharan, 2022 [107]	USA	2	Bilateral Nacc	Responsive pulses delivered bilaterally at 125 Hz in two 5 s bursts; charge density of 0.5 μC cm^−2^	BED and severe obesity	Failure of either 6 months of pharmacotherapy, 6 months of behavioral therapy, or gastric bypass therapy	BMI 45 to 60	BMI loss of −2.2 kg/m^2^ (−4.5%) in Subject 1, −2.9 kg/m^2^ (−5.8%) in Subject 2 at 6-month follow-up	Reduced loss-of-control eating frequency (Subject 1: 80% decrease; Subject 2: 87% decrease). No QoL or social functioning scales reported.

ALIC, anterior limbs of the internal capsule; AN, anorexia nervosa; BAI, Beck Anxiety Inventory; BDI, Beck Depression Inventory; BED, binge eating disorder; BITE, Bulimic Investigation Test Edinburgh; BNST, bed nucleus of the stria terminalis; BUT, Body Uneasiness Test; DERS, Dysfunction in Emotional Regulation Scale; EAT-26, Eating Attitudes Test, 26-item; EDE, Eating Disorder Examination; GAF, Global Assessment of Function scale; HAMA, Hamilton Anxiety Rating; HAMD, Hamilton Depression Rating; MADRS, Montgomery–Asberg Depression Rating Scale; MDD, major depressive disorder; MMSE, Mini-Mental State Examination; Nacc, nucleus accumbens; SCC, subcallosal cingulate; SHAPS, Snaith–Hamilton Pleasure Scale; SF-36, Short-form Health Survey; WHO-QoL-Psych, World Health Organization Quality-of-Life Assessment; YBC-EDS, Yale–Brown–Cornell Eating Disorders Scale; YBOCS, Yale–Brown Obsessive Compulsive Scale.

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
