# Peer review of "Neuromodulation of Eating Disorders: A Review of Underlying Neural Network Activity and Neuromodulatory Treatments"

_brainsci, 2024, doi:10.3390/brainsci14030200_

Round 1
Reviewer 1 Report
Comments and Suggestions for Authors
The authors present a thorough review of the current neuromodulation therapeutical options in the realm of eating disorders. The main eating disorders, AN, BN and BED are clearly presented for the un-initiated readers. The authors go into detailed presentations of the brain networks implicated in these disorders, based on neurophysiological and imaging research. In turn, they present non-invasive neuromodulation options, accompanied by data regarding efficacy of treatment, for each eating disorder. They also devote a significant portion of the review to more invasive approaches and productively discuss the clinical evidence so far. Finaly, they constructively outline their views on neuromodulation treatments and discuss potential future directions. The review is well structured and quite convenient for the reader to go through. I have no major comments and I am happy to endorse this excellent review for publication as is.
Author Response
Dear reviewer,
Thank you for taking the time to review this manuscript! We greatly appreciate your comments and thank you for your support of our manuscript.
Best regards,
Kevin Wu
Reviewer 2 Report
Comments and Suggestions for Authors
This review covered brain stimulation techniques for various eating disorders and the underlying neural mechanisms mediating eating disorders. After an Introduction, the review covered pathological alterations in various brain networks in several eating disorders. This section was concise, but detailed and very good. The main part of the review focused on rTMS and tDCS for eating disorders. Smaller sections were devoted to DBS and VNS.
Overall, this review was very well-written, was easy to understand, and had only a few grammatical or typographical errors. I think the review adds to the literature and provided a detailed summary of the literature on the topics of brain stimulation and eating disorders. The review will be of interest to readers of the journal and researchers is several related fields.
I only have a few minor comments of various types that the authors should consider and a few minor corrections to suggest.
1. My major comment is that the tables are very good, but the format make them very difficult to read with the spacing, large blank spaces, words cutting off and going to the next line etc. Is there anyway to improve this formatting?
2. My second major comment is it is a little unclear about what criteria was used to determine which studies were included in the review, especially for the rTMS and tDCS studies. Were all studies done on the topic included or only the best ones etc? Please give more info on this.
3. Bibliography: It appears that there are formatting inconsistencies in the bibliography. Most of the journal names are abbreviated but some are not for instance #s 9, 10, 19, 29, 60 etc.
4. There appear to be a few examples of too many spaces between sentences (e.g. Lines 291, 359, 472, 629). A little more proofreading needed.
5. Page 24 table, some columns are erroneously bolded
6. Line 590 “Investigations” shouldn’t be capitalized
Comments on the Quality of English Language
See comments to authors
Author Response
Dear reviewer,
Thank you for taking the time to review our manuscript. We greatly appreciate your feedback. Please see the attachment for a point-by-point response to your comments and suggestions.
Best regards,
Kevin Wu

Reviewer 3 Report
Comments and Suggestions for Authors
Dear Editor,
I have gone through the manuscript draft titled “Neuromodulation of Eating Disorders: A Review of Underlying Neural Network Activity and Neuromodulatory Treatments” that has been submitted to BS. The authors have done a commendable job of reviewing the neural networks involved in eating disorders, particularly anorexia nervosa, bulimia nervosa, and binge eating disorder. The review also provides a comprehensive summary of the available evidence on using non-invasive (rTMS, tDCS, and ECT) and invasive (DBS and VNS) treatment alternatives.
The manuscript draft is well-written and insightful. The appropriate tables, figures, and references support the manuscript, making it a valuable contribution to the field of eating disorders.
However, I do have some concerns regarding the study. Firstly, I would like to request further clarification on how the authors identified studies for their manuscript draft. Secondly, I would like to suggest that the authors consider including a critical appraisal of potential bias in the primary studies. Moreover, I think it would be helpful if all abbreviations were consistently used throughout the text. Finally, I noticed a few minor typos that the authors might want to address.
Overall, I believe that this review has the potential to significantly contribute to the field of eating disorders when published, and I am happy to recommend that the manuscript undergo a minor revision.
Thank you for considering my feedback.
Best regards.
Author Response

(The authors gave the same response as above.)
